# Trend and associated factors of cesarean section rate in Ethiopia: Evidence from 2000–2019 Ethiopia demographic and health survey data

Rahel Mezemir[1,2]*, Oladapo Olayemi[3], Yadeta Dessie[4]

**1** Pan African University, Life and Earth Sciences Institute (Including Health and Agriculture), Ibadan, Nigeria, **2** St. Paul's Hospital Millennium Medical College, School of Nursing, Addis Ababa, Ethiopia, **3** Department of Obstetrics and Gynaecology, College of Medicine, Pan African University Life and Earth Sciences Institutes, University of Ibadan, Ibadan, Nigeria, **4** College of Health and Medical Sciences Haramaya University, Harar, Ethiopia

* rahelmezemir7@gmail.com, rahel.mezemir@sphmmc.edu.et

## Abstract

### Background

The world health organization considers cesarean section (CS) prevalence of less than 5% suggests an unmet need. On the other hand, a prevalence of more than 15% may pose to risk to mother and child, however, access to CS in a resource-limited country like Ethiopia was much lower than the aforementioned level, Therefore, this was the first study to determine the trend of CS, and factors that influence it.

### Methods

This was done based on the five Ethiopia Demographic and Health Surveys. Trend analysis was done separately for rural and urban. The significance of the trend was assessed using the Extended Mantel-Haenszel chi-square test. The factors on CS delivery were identified based on DHS 2016 data. A multi-level logistic regression analysis technique was used to identify the factors associated with cesarean section delivery. The analysis was adjusted for the different individual- and community-level factors affecting cesarean section delivery. Data analysis was conducted using STATA 14.1 software.

### Result

The rate of cesarean section increased from 5.1% in 1995 to 16% in 2019 in an urban area and 0.001 in 1995 to 3% in a rural area, the overall increment of CS rate was 0.7% in 1995 to 2019 at 6%. The odds of cesarean section were higher among 25–34 years (AOR = 2.79; 95% CI: 1.92, 4.07) and 34–49 years (AOR = 5.23;95% CI: 2.85,9.59), among those educated at primary school level (AOR = 1.94; 95% CI: 1.23,3.11), secondary education (AOR = 2.01; 95% CI: 1.17, 3.56) and higher education (AOR = 4.12; 95% CI: 2.33–7.29) with multiple pregnancies (AOR = 11.12; 95% CI: 5.37, 23.), with obesity (AOR = 1.73; 95%

**Data Availability Statement:** All datasets and material used in the present study that support the findings are available in the paper.

**Funding:** The authors received no specific funding for this work.

**Competing interests:** The authors have declared that no competing interests exist.

**Abbreviations:** AOR, Adjusted Odds Ratio; CI, Confidence Interval; CS, Caesarian Section; EDHS, Ethiopia Demographic and Health Survey; EOC, emergency and obstetric care; ICC, Intra Class Correlation Coefficient; LLR, Log-Likelihood Ratio; MOR, Median odds ratio; OBGY, obstetrics and gynecology; OR, Odds Ratio; PVC, Proportional Change in Variance; SNNPR, Southern Nations, Nationalities, and Peoples' Region.

CI: 1.22, 2.45), living in an urban area (AOR = 2.28; /95% CI: 1.35–3.88), and increased with the number of ANC visit of 1–3 and 4th(AOR = 2.26; 95% CI: 1.12, 4.58), (AOR = 3.34; 95% CI: 1.12, 4.58), respectively. The odds of cesarean section are lower among parity of 2–4 children (AOR = 0.54; 95% CI: 0 .37, 0.80) and greater than four birth order (AOR = 0.42;95% CI: 0.21,0.84).

## Conclusion

In Ethiopia, the CS rate is below the WHO recommended level in both urban and rural areas, thus, intervention efforts need to be prioritized for women living in a rural area, empowering women's education, encouraging co-services such as ANC usage could all help to address the current problem.

## Introduction

In obstetrical care, Cesarean section (CS) is the most routinely performed surgery, it is the delivery of a fetus through a surgical incision in a pregnantwomen's abdominal wall and uterine wall. It is commonly done for maternal or fetal reasons to avert maternal and neonatal morbidity and mortality rate [1]. Globally, the use of CS among delivery women have continuously increased, and it is predicted to continue to rise over the next decade, in both developed and developing countries [2] although the availability of safe CS in the resource-limited country is still much lower [3]. Recent data from 154 countries show that the global average CS rate of 21.1%, with averages of 8.2% at least, 24.2% in less, and 27.2% in more developed countries. Sub-Saharan Africa has the lowest rates (5.0 percent, 39 nations, 88.6 percent birth coverage), while, Latin America and the Caribbean have the highest rates (42.8 percent, 23 countries, 91.2 percent birth coverage [3]. The world health organization considers (CS) prevalence of less than 5% suggests an unmet need. On the other hand, a prevalence of more than 15% may pose to risk to mother and child [4]. Reports from the population-based study also support this finding CS rates greater than roughly 10% in the population are not associated with lower maternal and neonatal mortality rates [5]. Another ecological study finds a negative and statistically significant linear correlation between CS rate (the number of CS deliveries per 100 births) and neonatal mortality [6]. Despite all of these facts, in the recent years, the medically non-indicted CS section is significantly practiced [7]. Evidence from the WHO survey shows this worldwide increase in medically non-indicated CS can have a substantial impact on awomen's health and medical costs, like, increased rate of infection, longer healing time, significant bleeding, increased iatrogenic injury, increased breathing difficulties, and even death [8, 9]. Similarly, a multi-country facility-based survey came to the same conclusion [2]. Several studies show that medically and non-medically indicated factors including the type of health institution, socio-demographic characteristics, and maternal health of women, have been linked to an increase in CS. Some of these factors are the mother's age, birth order, place of residence, socioeconomic status, maternal educational level, previous CS, obstetric problems, maternal request, and income level [9–11]. These variables also differ depending on the population [12]. In Ethiopia, research from the available resource on the prevalence of CS indicates a figure well below the WHO's recommended threshold of 10% [4]. Furthermore, for numerous years, there has been no notable increase in the country's population-based CS rates. for example, in the 2016 and 2000 Ethiopia Demographic and Health Surveys (EDHS), national

CS rates were found to be very low (1.9 percent and 0.7 percent, respectively) [13, 14]. This figure is lower than that of several African countries, including Ghana (13% in 2014) [15], Nigeria (2.1% in 2013) [16], and Mozambique (4.7 in 2014) [17]. The relatively low prevalence of CS in the Ethiopian population reflects unmet demands, which may contribute to the country's poor maternal and newborn outcomes [18]. Evidence from hospital-based and population-based cross-sectional studies in Ethiopia shows, delivery by CS most common in women aged 35–49, first-parity births, births for which women had more than three Antenatal Care (ANC) visits, births in urban areas, higher education, and higher economic status (15%) [10, 19, 20]. In Ethiopia, most study on CS section was hospital base data, this type of data was most of time used small sample size and also subject to selection bias. Few studies were used population based data, however, the studies had the some limitation, for instance, the study was not used higher statistical software for analysis like multilevel analysis, which used to identify individual and community level factors.

Using a nationally representative sample has several advantages over other institutional sample data in terms of calculating the compiled trend to highlight national-level characteristics, which could lead to policy decisions at the national level. Therefore, the present study used to assess the trend of CS in the urban and rural area factors that influencing it.

## Methods and material

### Study setting

The study was done in Ethiopia. Ethiopia is a multi-ethnic country in east Africa with a diverse population. It is bordered on the west by Sudan, on the east by Somalia and Djibouti, on the north by Eritrea, and the south by Kenya. The country has a total area of 1,112,000 square kilometers

Ethiopia is divided into eleven regions and two municipal governments. Tigray, Afar, Amhara, Oromiya, Somali, Benishangul-Gumuz, Southern Nations Nationalities and People (SNNP), Gambela, Sidama, South Western, and Harari are among the regions involved. Addis Ababa and Dire Dawa are administrative cities [21]. According to the EDHS 2016 and 2011, the prevalence of CS in Ethiopia was 1.9% [13] and 0.7% [22], respectively [13, 22].

### Study design

The research was based on secondary data from the EDHS. Factors associated with CS were identified using EDHS 2016 data, whereas trend analysis was done using EDHS 2000, 2005, 2011, 2016, and 2019 (mini EDHS) data. Since EDHS collected data about births in the previous 5 years, the data indicated CS from 1995 to 2019 [13].

### Sample size and sampling methods

All EDHS surveys used a sample that was aimed to represent all of the country's regions and administrative cities. The survey participants were chosen using a two-stage stratified sampling technique. The first stage was a selection of the enumeration areas. The enumeration areas were stratified into urban and rural. In the second stage, households in the selected enumeration area were selected. The sample size was then allocated using a probability-proportional allocation method. 645 enumeration areas (EAs) were chosen for the 2016 DHS. There were 202 EAs from urban regions and 443 from rural areas. Six hundred twenty-four EAs were included in the 2011 DHHS (187 from urban regions and 437 from rural areas) [13]. The 2005 EDHS had 540 EAs (145 from urban areas and 395 from rural areas), while the 2000 DHS included 539 EAs (138 from urban and 401 from rural) [14, 22]. Then, on average 27 to 32

households per EA were selected from all surveys. The source population was all live births from reproductive-age women within 5 years before the survey in Ethiopia. A total weighted sample of 46,317 live births (12,260 in EDHS 2000, 11,163 in EDHS 2005, 11,872 in EDHS 2011, and 11,022 in EDHS 2016) was used for analysis. Detailed sampling procedure can be found from the EDHS [13, 14, 22, 23].

## Data collection

Five interviewer-administered questions were used by the EDHS: the household questionnaire, thewomen questionnaire, the men questionnaire, the biomarker questionnaire, and the health facility questionnaire [4, 14, 22, 23]. Data was collected for this study from children under the age of five surveys, born to interviewed mothers who gave birth within five years of the survey year 1995–2016, which was included in the kid records. The data collection tool was created in English initially, then translated into the country's three main languages: Amharic, Oromiffa, and Tigrigna. The Somaligna and Afarigna languages were also used in the 2000 DHS [4, 22].

## Variable of study

**Outcome variable.** The outcome variable in this study is the CS which was taken dichotomous and coded by the value "1" (one) if the respondents underwent cesarean delivery and "0" (zero) if not.

**Independent variable.** There were three categories of independent variables; institution-related, socio-demographic and economic factors, and pregnancy-related factors. Institutional factors include the place of delivery (public vs private), the number of antenatal care visits (no visit, 1–3 and >4), pregnancy-related factors including parity (Primi-parous, multi-parous, and Grand-multi-parous), birth order (first, second, third or higher), maternal, body–mass index (normal, underweight and overweight), Size of the baby was determined from the maternal recall of baby's weight at birth (very large, average, smaller than the average), socio-demographic and economic factors consist of maternal education, maternal age at birth, marital status, mothers' employment status (yes/no), wealth index (poor, middle, rich), residences, and region.

**Statistical analysis.** Completed EDHS questionnaires *were* meticulously tagged, entered, and modified after data collection The distribution of study participants in the sample was weighted to create nationally representative data [22]. STATA software version 14 was used to analyze the data. Frequency and percentage were utilized as descriptive statistics. Using chi-square analysis, the CS rate was compared across several socio-economic, maternal, and child characteristics. The DHS surveys gathered information on the mode of delivery of birth within the previous five years. The rate was calculated for each year between 1995 and 2019 based on the specific year of delivery, 2019 mini DHS data was included for the trend analysis, however, for determinate factors, the data was not completed.

The Extended Mantel-Haenszel chi-square test for linear trend was used to examine the significance of the trend of the CS rate using the OpenEpi software (Version 3.01) dose-response program [24]. A 95% significant probability of the existence of a trend was declared when the p-value was less than 0.05. Further, the change in trend CS rate is presented in two ways, Absolute increase of CS rate and relative increase as the average annual rate of increase (AARI), to find the absolute change increase, subtract the latest CS rate from the earliest CS rate and to find an average annual rate of increase, AARI = [(an / am) [1 / (n-m)]]-1; where am; is the first observation of CS rate, and; is the latest observation of CS rate, m is the first observed year and n is the latest observed year. The AARI is a geometric progression ratio that provides a constant rate of change during the study period [3].

To identify factors associated with CS delivery, a Multi-level logistic regression analysis technique was applied, since the data had hierarchical and clustering nature.

A total of four models were carried out. The first model was an empty model that was used to calculate the random variability in the intercept. The second model estimated the influence of individual-level factors on CS delivery. The third model looked at how community-level factors are associated with CS delivery. Finally, the fourth model computed the influence of individual and community-level factors on cesarean delivery. The Intra-Cluster Correlation (ICC) was determined to illustrate the correlation between clusters within a model, and the intra-cluster correlation (ICC) is expected to be $\geq 10\%$ when using this model. The power of variables included in each model in predicting CS delivery was also determined using the Proportional Change in Variance (PCV). To determine the factors that associated with cesarean section, the model with the highest PCV value was used. Significant factors were considered as variables with a p-value less than 0.05.

**Ethical consideration.** All Ethiopian Demographic and Health Surveys obtained ethical approval from the Ethiopian Health and Nutrition Research Institute Review Board, the Ministry of Science and Technology, ICF International's Institutional Review Board, and the CDC. Data was collected after informed consent was obtained, and all information was kept private. After reviewing the brief descriptions of the study provided to the DHS program, the Demographic, and Health Surveys Program granted authorization to access EDHS data for this specific research. The data sets were handled with the utmost confidentiality [13].

## Result

### Socio-demographic and economic characteristics of the study population

In the four EDHS, information on cesarean Section deliveries were collected from 46,316 women who had given birth in the five years before the survey period. The median ages (+IQR) of women in the four surveys were similar with the most recent (2016 DHS) being 26.1 + 9 years. In all four surveys, three -fourth of live birth were from rural areas, the latest report (2016 DHS) was 88.8% and most of the live births were from the Oromia region. On the other hand, the percentage of women having primary education increased slightly from 13% in 2000 to 26.8% in 2016. In all four surveys, the majority of women were married (Table 1).

### Child characteristics and maternal health service usage

The median BMI of women were similar across the four surveys, the recent one (DHS 2016) was 21.1kg/m$^2$ with a range between 53.6 kg/m$^2$ and 12.3kg/m$^2$. Almost one-fifth of women had one to three parity across the four surveys, the highest was in (DHS2011), 44.6%. Furthermore, of all live birth born from the interviewed women, more than half of them were single birth across four surveys. Among the birth order, fourth or higher-order births accounted for about half of all births (51.1%). The percentage of women giving birth at home decreased from 94.9% (in the first survey) to 72.2% (in the fourth survey). The percentage of women who had one to four ANC visits increased from 16.4% in 2000 to 30.9% in 2016 (Table 2).

### The magnitude of the cesarean section rate

CS rate in the urban area was (5.1% in 2000, 9.4% in 2005), and 10.6% (in 2016). Furthermore, in all four survey years, the percentage of CS among women of overweight (from 7.6% in 2000 to 7.8% in 2016) with women of > 5 parity (from 0.1% in 2000 to 0.4% in 2016). The number of ANC visits > 4 (from 1.8% in 2000 to 7.5% in 2016) (Table 3). The rate of CS delivery among women in the age group of 25–34 and 35–49 years (0.3% in 2000 to 1.9 in 2016 and 0.1

**Table 1. Sociodemographic and economic characteristics of the study population, finding from 2000 to 2016 DHS.**

| Variable | Category | Live birth in 2000DHS (n = 12,260) | Live birth in 2005DHS (n = 11,163) | Live birth in 2011DHS (n = 11,872) | Live birth in 2016DHS (n = 11,022) |
|---|---|---|---|---|---|
| | | (%) | (%) | (%) | (%) |
| Mother's age at birth | Median +IQR | 26 ±10 | 26 ±10 | 26±9 | 26±9 |
| Regions/ administrations | Tigray | 788(6.4) | 698 (6.3) | 753 (6.3) | 716 (6.5) |
| | Afar | 126(1.0) | 107 (1.0) | 121 (1.0) | 114 (1.0) |
| | Amhara | 3202(26.2) | 2621 (23.5) | 2656 (22.4) | 2072 (18.8) |
| | Oromiya | 4999(40.8) | 4411 (39.5) | 5014 (42.2) | 4851 (44.2) |
| | Somali | 142(1.1) | 477 (4.3) | 364 (3.1) | 507 (4.6) |
| | Ben-Gumz | 124(1.0) | 105 (0.9) | 140 (1.2) | 122 (1.1) |
| | SNNP | 2602(21.2) | 2500 (22.4) | 2494 (21.1) | 2296 (20.8) |
| | Gambela | 29(0.2) | 31 (0.3) | 40 (0.3) | 27 (0.2) |
| | Harari | 25(0.2) | 22 (0.2) | 29 (0.2) | 26 (0.2) |
| | Addis | 182(1.5) | 153 (1.3) | 221 (1.9) | 244 (2.2) |
| | Dire Dawa | 40(0.3) | 37 (0.3) | 39 (0.3) | 47 (0.4) |
| Residence | Urban | 1277(10.4%) | 815(7.3) | 1,528(12.9) | 1,216(11.0) |
| | Rural | 10,984(89.5) | 10,348(92.7) | 10,344(87.1) | 9,807(88.8) |
| Highest education | Illiterate | 10,063(82.1) | 8838(79.1) | 8,227(69.3) | 7,284(66.1) |
| | Primary | 1,596(13.1) | 1845(16.2) | 3,211(27.1) | 2,951(26.8) |
| | Secondary | 573(4.7) | 427(3.8) | 266(2.2) | 514(4.7) |
| | Higher | 28(0.2) | 43(0.3) | 168(1.4) | 274(2.5) |
| Marital status | Never married | 64(0.5) | 37(0.3) | 78(0.7) | 57(0.5) |
| | Married | 11,270(91.9) | 10,518(94.2) | 10,989 (92.6) | 10,462(94.9) |
| | Widowed | 189(1.5) | 183(1.6) | 229(1.9) | 384(3.5) |
| | Divorced | 738(6.1) | 424(3.8) | 576(4.8) | 274 (2.5) |
| Employment status | Yes | 6,858(55.9) | 2,590(23.2) | 4,060(34.2) | 8,035(72.9) |
| | No | 5,397(44.1) | 8,572(6.8) | 7,803 (65.7) | 2,988(27.1) |
| Wealth quantile | Poorest | NA | 2,439 (21.9) | 2,709(22.8) | 2,636(23.9) |
| | Poorer | NA | 2,356(21.1) | 2,658(22.4) | 2,519(22.9) |
| | Middle | NA | 2,486 (22.3) | 2,437(20.5) | 2,279(20.7) |
| | Richer | NA | 2,223(19.9) | 2,272(19.14) | 1,999(18.1) |
| | Richest | NA | 1,659(14.9) | 1,795(15.1) | 1,588(14.4) |

IQR-Interquartile range

in 2000 to 1.8 in 2016) survey period respectively. The CS rate in private health facilities was 21.7% (Fig 1).

## The trend of cesarean section

There was a significant variation in the trend of the rate of CS observed in the urban and rural area in five DHS data (the 2019 mini DHS data was included). According to this report, the trend of CS in an urban area over the preceding 5 years of the surveys had increased from 5.1% (95% CI: 3.7–7.1) in 2000, 9.4% (95% CI: 6.5,13.4) in 2005, 8.1% (95% CI: 6.3,10.5) in 2011, 10.6. (95% CI: 8.5, 13.1) in 2016 and 10.1(95% CI: (7.2, 14.0) in 2019 although there is an over-lapping of the confidence interval in the year 2000 and 2005 and also 2011 and 2016. Similarly, the trend of CS in rural areas had increased from 0.2% (95% CI: 0.1–0.3) in 2000, 0.3% (95%

**Table 2. Child and maternal, characteristics and health service usage finding from 2000 to 2016 DHS.**

| Variable | Category | Live birth in 2000DHS (n = 12,260) | Live birth in 2005DHS (n = 11,163) | Live birth in 2011DHS (n = 11,872) | Live birth in 2016DHS (n = 11,022) |
|---|---|---|---|---|---|
| | | (%) | (%) | (%) | (%) |
| BMI | Median | 19.8 | 20.1±21.6–18.6 | 19.9+21.8–18.5 | 20.1+53.6–12.29 |
| Parity | 1 | 1,362 (11.1) | 4576 (41.0) | 5295 (44.6) | 4836 (43.9) |
| | 2–4 | 4038(32.9) | 3962 (35.5) | 4014 (33.8) | 3732 (33.9) |
| | >5 | 2884(23.5) | 2625 (23.5) | 2564 (21.6) | 2454 (22.2) |
| Size of the baby | Very large | 3,769(30.7) | 3557(31.9) | 3,805(32.1) | 3,485(31.6) |
| | Average | 4376(35.7) | 4462(39.9) | 4,547(38.3) | 4,580(41.6) |
| | Very small | 4035(33.3) | 3096(27.7) | 3,468.5(29.2) | 2,866(26.0) |
| Birth order | 1–2 | 4353(35.5) | 3673(32.9) | 4,279 (6.1) | 3,842(34.8) |
| | 3 | 1,668(13.6) | 1611(14.4) | 1,677(14.1) | 1,575(14.3) |
| | 4+ | 6,239(50.9) | 5879(52.6) | 5,916(49.8) | 5,605(50.9) |
| Child is twins | Singleton | 11,994(97.8) | 10,963 (98.2) | 11,597 (97.7) | 10,730 (97.4) |
| | Multiple | 266(2.2) | 200 (1.8) | 275 (2.3) | 292 (2.6) |
| Place of delivery | Home | 11,625(94.9) | 10,502(94.0) | 10,627(89.5) | 7,997(72.6) |
| | Public sector | 575(4.7) | 529.4(4.7) | 1,005(8.5) | 2,734(24.8) |
| | Private sector | 40(0.3) | 54(0.5) | 144(1.2) | 157(1.4) |
| | Others | 11(0.1) | 45(0.4) | 50(0.42) | 134(1.2) |
| Number of ANC visit | No visit | 5,789(72.6) | 5225(71.5) | 4,517 (57.1) | 2,818(37.1) |
| | 1–3 | 1,309(16.4) | 1163(15.9) | 1,856(23.5) | \|2,342(30.9) |
| | ≥4 | 831(10.4) | 888(12.1) | 1,508(19.01) | 2,415(31.8) |

BMI- body mass index, ANC–antenatal care

CI: 0.2, 0.6) in 2005, 0.5% (95% CI: 0.3, 0.8) in 2011, 0.9% (95% CI: 0.3,1.6) in 2016 and 3.9% (95% CI2.8,5.4) in 2019. Based on the chi-square test for linear trend, the increment was statically significant in urban ($X^2 = 20.72$, p-value<0.0001) and rural ($X^2 = 52.72$, p-value<0.0001) areas respectively.

In addition to this, the rate sustainably increased among administration regions. For example, Addis Ababa had the highest percentage of (24.1%) in 2019 and the biggest growth from the year 2000 (7.9%). The rate was statistically significant at (p<0.0001) (Fig 2).

Fig 3 Presents a year-specific trend of CS rate, survey data illustrated the rate of CS in urban had risen from 5.8% (in 1995) to 16% (in 2019). Similarly, the rate of CS in rural had risen from 0.001% (in 1995) to 3% (in 2019) since there is no CS delivery in 1995 in the rural part. The overall increment in CS rate was also observed from 0.8% (1995) to 6% (in 2019). Based on the chi-square linear trend, the rate was statistically significant at (p<0.0001), (p<0.0001), and (p<0.0001), in urban, rural, and overall respectively.

Furthermore, the absolute changes of CS rate in urban areas from (1995–2019) was 10.2% whereas, the average annual rate of increased (AARI) from (1995–2019) was 4%. In rural area from (1995–2019), 2.9% and 6.8% was the absolute change and AARI respectively. The overall CS rate increment from (1995–2019) was 2.2% absolute change and 6.7% AARI.

## Factors associated with cesarean section rate

Table 4 presents the findings of a multi-level logistic regression,the study evaluating the relationship between CS and a variety of individual factors and community-level contextual

**Table 3. The magnitude of cesarean section rate across the various characteristics of the respondents in Ethiopia, finding from 2000 to 2016.**

| Variable | Category | Live birth in 2000DHS (n = 12,260) | | Live birth in 2005DHS (n = 11,163) | | Live birth in 2011DHS (n = 11,872) | | Live birth in 2016DHS (n = 11,022) | |
|---|---|---|---|---|---|---|---|---|---|
| | | (%) | CI | (%) | CI | (%) | CI | (%) | CI |
| Age | 18–24 | 1.3 | 0.9,1.9 | 1.2 | 0.8,1.7 | 1.9 | 1.4,2.6 | 1.7 | 1.2,2.5 |
| | 25–34 | 0.3 | 0.2,0.5 | 1 | 0.7,1.5 | 1.1 | 0.8,1.5 | 2.2 | 1.7,2.8 |
| | 35–49 | 0.1 | 0.0,0.2 | 0.5 | 0.2,1.2 | 1.3 | 0.6,2.9 | 1.9 | 1.1,3.2 |
| Residence | Urban | 5.1 | 3.7,7.1 | 9.4 | 6.5,13.4 | 8.1 | 6.3,10.5 | 10.6 | 8.5,13.1 |
| | Rural | 0.2 | 0.1,0.3 | 0.3 | 0.2,0.6 | 0.5 | 0.3,0.8 | 0.9 | 0.6,1.3 |
| Highest education | Illiterate | 0.1 | 0.1,0.2 | 0.4 | 0.2,0.6 | 0.4 | 0.2,0.7 | 0.7 | 0.4,1.1 |
| | Primary | 1.2 | 0.6,2.2 | 0.9 | 0.5,1.6 | 2.4 | 1.5,3 | 2.5 | 1.8,3.6 |
| | Secondary | 8.4 | 5.1,13.6 | 12.1 | 8.5,16.8 | 14.6 | 10.0,21.0 | 6.3 | 4.2,9.5 |
| | Higher | 18.8 | 7.1,41.5 | 23 | 11.7,40.1 | 13.7 | 8.3,21.9 | 20.8 | 15.7,26.9 |
| Marital status | Never married | 3.9 | 0.6,20.5 | 4.2 | 0.6,23.2 | 7.2 | 2.3,20.1 | 14.2 | 6.4,28.7 |
| | Married | 0.5 | 0.3,0.7 | 0.9 | 0.7,1.2 | 1.4 | 1.1,1.9 | 1.8 | 1.5,2.3 |
| | Widowed | 2.5 | 0.4,12.6 | 1.3 | 0.3,6.5 | 1.2 | 0.4,4.3 | 0.5 | 0.1,3.7 |
| | Divorced | 3.2 | 1.3,7.8 | 1.9 | 0.8,4.2 | 1.8 | 0.9,3.7 | 3 | 1.5,5.8 |
| Employment status | Yes | 0.7 | 0.4,1.2 | 1.6 | 1.0,2.5 | 1.7 | 1.2,2.3 | 3.1 | 2.4,4.2 |
| | No | 0.7 | 0.4,1.1 | 0.8 | 0.6,1.1 | 1.4 | 1.0,2.0 | 1.5 | 1.1,2.0 |
| Wealth quantile | Poorest | NA | NA | 0 | 0.0,0.2 | 7.2 | 2.3,20.1 | 0.6 | 0.2,1.4 |
| | Poorer | NA | NA | 0.3 | 0.1,0.8 | 1.4 | 1.1,1.9 | 1 | 0.4,2.4 |
| | Middle | NA | NA | 0.2 | 0.1,0.5 | 1.2 | 0.4,4.3 | 1 | 0.5,2.0 |
| | Richer | NA | NA | 0.5 | 0.2,1.0 | 1.8 | 0.9,3.7 | 1 | 0.5,1.9 |
| | Richest | NA | NA | 5.3 | 3.9,7.2 | 7.2 | 2.3,20.1 | 8.1 | 6.6,9.9 |
| BMI | normal (18.05–24.09) | 0.5 | 0.3,1.0 | 0.7 | 0.4,1.2 | 1 | 0.7,1.3 | 1.6 | 1.2,2.1 |
| | Underweight (<18.04) | 0.5 | 0.3,0.8 | 0.6 | 0.2,1.7 | 0.9 | 0.4,2.0 | 0.9 | 0.3,2.5 |
| | overweight (>25.00) | 7.6 | 3.1,17.5 | 8 | 3.9,16.0 | 8.6 | 5.8,12.5 | 7.8 | 5.5,11.1 |
| Parity | 1–3 | 1.5 | 1.1,2.1 | 2 | 1.5,2.7 | 2.7 | 2.1,3.3 | 3.7 | 3.0,4.5 |
| | 4–6 | 0.1 | 0.0,0.3 | 0.1 | 0.1,0.4 | 0.7 | 0.3,1.5 | 0.7 | 0.4,1.3 |
| | >6 | 0.1 | 0.0,0.3 | 0.5 | 0.2,1.1 | 0.2 | 0.1,0.6 | 0.4 | 0.2,1.0 |
| Size of the baby | Very large | 1 | 0.6,1.5 | 1.3 | 0.9,2.0 | 1.7 | 1.2,2.5 | 2.6 | 1.9,3.5 |
| | Average | 0.7 | 0.4,1.1 | 0.9 | 0.6,1.4 | 1.6 | 1.0,2.6 | 1.7 | 1.2,2.4 |
| | Very small | 0.5 | 0.3,0.9 | 0.7 | 0.4,1.2 | 1 | 0.7,1.5 | 1.5 | 1.0,2.2 |
| birth order | 1–2 | 1.8 | 1.3,2.6 | 2.4 | 1.8,3.2 | 2.9 | 2.3,3.7 | 3.7 | 2.9,4.7 |
| | 3 | 0.1 | 0.0,0.3 | 0.4 | 0.2,0.8 | 1.1 | 0.6,1.9 | 2.5 | 1.6,3.8 |
| | 4+ | 0.1 | 0.0,0.2 | 0.3 | 0.1,0.6 | 0.6 | 0.3,1.1 | 0.6 | 0.3,0.9 |
| Child is twin | Singleton | 0.7 | 0.5,0.9 | 1 | 0.7,1.3 | 1.4 | 1.1,1.8 | 1.8 | 1.4,2.2 |
| | Multiple | 1.8 | 0.3,11.9 | 3.3 | 1.2,8.8 | 4.1 | 1.3,12.5 | 7.5 | 3.4,15.8 |
| Place of delivery | Public sector | 14.4 | 10.7,19.1 | 18.9 | 15.1,23.5 | 14.6 | 11.5,18.3 | 6.5 | 5.3,8.1 |
| | Private sector | 7.9 | 3.0,19.3 | 20 | 8.4,40.4 | 20.1 | 14.0,28.1 | 21.7 | 15.2,29.9 |
| ANC visit | No visit | 0.2 | 0.1,0.4 | 0.2 | 0.1,0.4 | 0.2 | 0.1,0.5 | 0.5 | 0.2,1.2 |
| | 1–3 | 1 | 0.4,2.6 | 1.5 | 0.7,2.9 | 2.4 | 1.5,3.8 | 1.3 | 0.8,2.2 |
| | ≥4 | 3.8 | 2.2,6.4 | 6.3 | 4.6,8.5 | 6 | 4.5,7.8 | 5.7 | 4.6,7.0 |

BMI- body mass index, ANC–antenatal care

factors (ModelsI-IV) using the recent 2016 DHS data. The association between the outcome and explanatory variables examined by using bivariate regression analysis. Variables were included in the multivariate multi-level regression analysis based on their association with the bivariate level. Model I is an empty model, on the other hand, model II and III show the

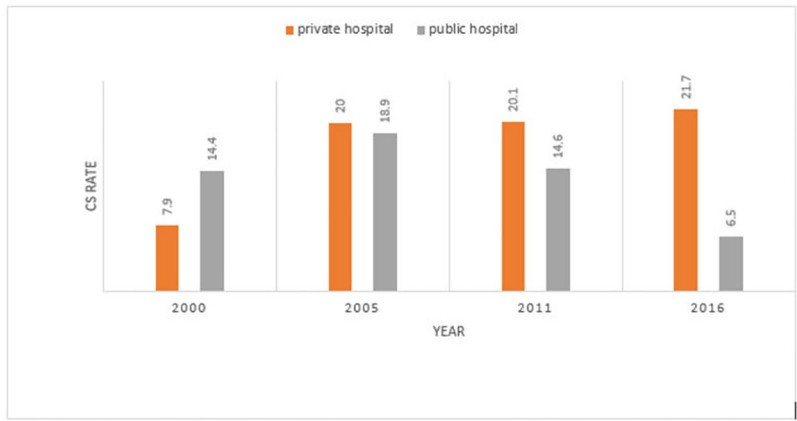

**Fig 1. Magnitude of CS rate among private hospital, public hospital (2000–2016) survey year.**

**Fig 2. The trend of CS rate across the administrative region from (2000–20019) survey year.**

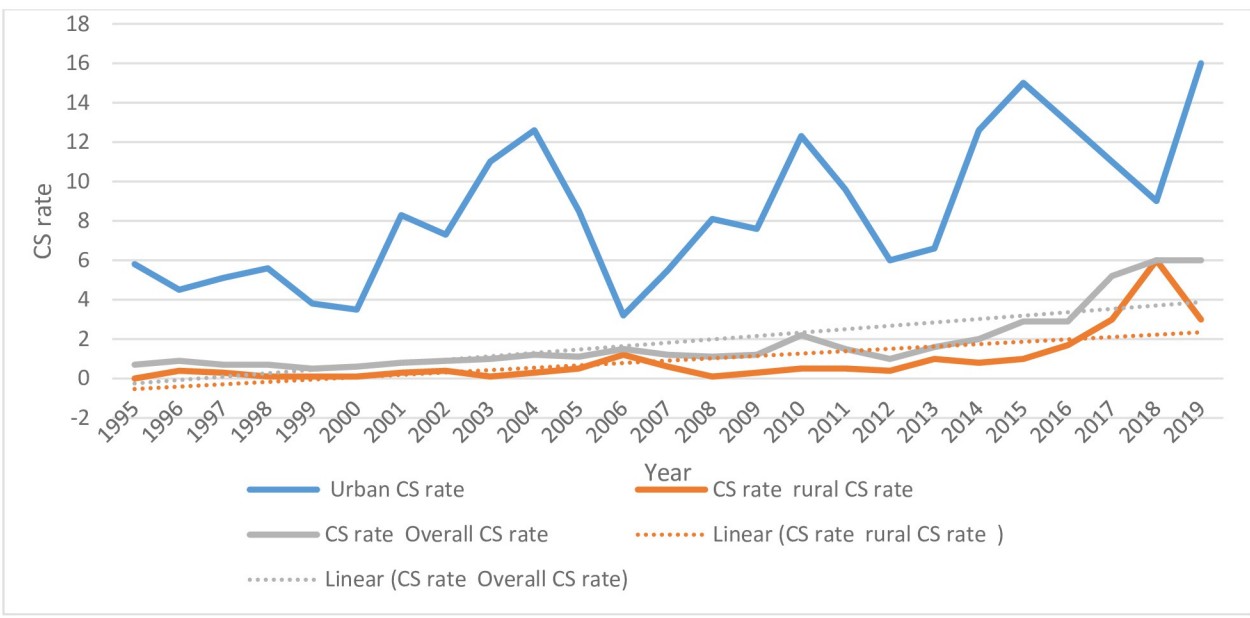

**Fig 3. The trend of CS rate in the Urban and rural part of Ethiopia data from (1995–2019).**

findings of individual and community-level variables, whereas model IV presents the results of all variables in the multilevel regression model. The ICC in Model I (empty model) revealed 63.9% variability in CS delivery which related to community variations (between-cluster variability). Similarly, the between-cluster variability in Model II 11.9%, in Model III, and Model IV were 23.7% and 5.8%, respectively. According to the Proportional Change in Variance (PCV's) finding shows, the adding predictor variables to the empty model improved the explanation of factors related to CS delivery. Model II (individual-level variables) had PCV of 92.3%, Model III (community-level factors) had 82.5% PCV and Model IV (combined individual and community-level factors) had PCV of 96.6%. Model IV reveals that individual and community-level characteristics account for over 97% of the variability in CS delivery between communities, so it is better to identify the factors related to CS birth. Therefore, variables like the mother's age at birth, education, BMI, parity, birth order, ANC visit, type of pregnancy, residence, and geographical region were statistically associated with CS delivery. From the final model, the odds of having CS delivery by women age group 34–49 years are 5.2 times more likely compared to those age group of 18–24 years (AOR = 5.23; 95% CI:2.85, 9.59, p-value <0.000). Those secondary and higher education increased the odds of CS delivery by 2 and 4.1 times than no education (AOR = 2.01;95% CI:1.17, 3.56, p-value <0.012) and (AOR = 4.12;95% CI:2.33–7.29, p-value <0.000) respectively. Women who gave birth multiple pregnancies have increased the odds of CS delivery by 11.2 times as compared to that singleton (AOR = 11.12; 95% CI:5.37, 23.23, p-value < 0.000). The odds of CS decline with increasing parity and birth order, that means women with 2–4 children CS lower by 46% as compared to one child (AOR = 0.54; 95% CI:0 .37, 0.80, p-value < 0.002) and women with a birth order of greater than 4, the odds of having CS decline by 58% as compared to those with a birth order of one (AOR = 0.42;95% CI:0.21,0.84, p-value <0.023). Furthermore, women with a body mass index greater than 25 the odds of delivery by CS were increased by 1.7 times than women who had normal body mass index (AOR = 1.73;95% CI:1.22,2.45, p-value <0.002). Women who had one to three and four or more ANC visits increased the odds of CS by 2.2 and 3.3

**Table 4.** **Multilevel multivariable logistic regression output for cesarean section rate among live births in Ethiopia 2016.**

| Variable | Null model | Model II AOR (95% CI) | Model II AOR (95% CI) | Model IV AOR (95% CI) |
|---|---|---|---|---|
| Mother's age at birth | | | | |
| 18–24 | | | | |
| 25–34 | | 2.79(1.92–4.06)*** | | 2.79(1.921–4.07)*** |
| 35–49 | | 5.22 (2.85–9.59)*** | | 5.23(2.85–9.59)*** |
| Highest educational level | | | | |
| Illiterate | | | | |
| Primary | | 2.29(1.45–3.63)*** | | 1.92(1.23–3.11)** |
| Secondary | | 2.57(1.49–4.41)** | | 2.012(1.17–3.56)* |
| Higher | | 5.43(3.07–9.59)*** | | 4.12(2.33–7.29)*** |
| Marital Status | | | | |
| Never married | | | | |
| Married | | 0.50 (0.19–1.33) | | 0.58(0.23–1.55) |
| Widowed | | 0 .12(.01–1.21) | | 0.16(0.016–1.64) |
| Divorced | | 0.45(.15–1.35) | | 0.55(0.18–1.64) |
| Wealth quantile | | | | |
| poor | | | | |
| Middle | | 1.11 (0.56–2.21) | | 1.04(0.53–2.079) |
| Rich | | 2.36(1.46–3.84)*** | | 1.16(0.65–2.053) |
| BMI | | | | |
| normal (18.05–24.09) | | | | |
| Underweight (<18.04) | | 0.47(0.25–0.88)* | | 0.53(0.28–1.00) |
| overweight (>25.00) | | 2.08(1.49–2.93*** | | 1.73(1.22–2.45)** |
| Parity | | | | |
| 1 | | | | |
| 2–4 | | 0.53 (0.36–0.79)** | | 0.55(0.37–0.80)** |
| >5 | | 0.41(0.18–0.94) | | 0.45(0.19–1.03) |
| birth order | | | | |
| 1–2 | | | | |
| 3 | | 0.52(0.31–0.89)* | | 0.50(0.35–0.99) |
| 4+ | | 0.33(0.17–0.66)** | | 0.42(0.21–0.84)* |
| Child is twin | | | | |
| Singleton | | | | |
| Multiple | | 12.38(6.04–25.34)*** | | 11.16(5.37–23.23)*** |
| Number of ANC visit | | | | |
| No visit | | | | |
| 1–3 | | 2.71(1.34–5.46)** | | 2.26(1.12–4.58) |
| ≥4 | | 4.23(2.15–8.33)*** | | 3.34(1.67–6.66)** |
| Residence | | | | |
| Urban | | | 10.19(6.58–15.80)*** | 2.28(1.35–3.88)** |
| Rural | | | | |
| Region | | | | |
| Tigray | | | | |
| Affar | | | 0.28(0.09–0.81)* | 0.71(0.26–1.96) |
| Amhara | | | 1.01(0.47–2.22) | 0.96(.45–2.04) |
| Oromiya | | | 0.56(0.25–1.25) | 0.92(0.43–1.99) |
| Somali | | | 0.11(.04–0.33)*** | 0.39(0.12–1.25) |
| Ben-Gumz | | | 0.45(0.16–1.24) | 0.610(.21–1.72) |

*(Continued)*

**Table 4.** (Continued)

| Variable | Null model | Model II AOR (95% CI) | Model II AOR (95% CI) | Model IV AOR (95% CI) |
|---|---|---|---|---|
| SNNP | | | 1.30(0.63–2.72) | 1.24(0.62–2.47) |
| Gambela | | | 0.40(0.16–1.04) | 0.37(0.13–0.99) |
| Harari | | | 3.11(1.53–6.33)** | 3.00(1.55–5.81)** |
| Addis Ababa | | | 3.14(1.56–6.30)** | 1.51(0.81–2.81) |
| Dire Dawa | | | 1.14(0.524–2.46) | 1.73(0.87–3.38) |
| Random effect | | | | |
| Community variance(SE) | 5.83(0.93) | 0.45(0 .22) | 1.02(0.25) | 0.21(0 .21) |
| ICC | 63.9% | 11.99% | 23.73% | 5.76% |
| PCV | Reference | 92.32 | 82.46 | 96.55 |
| Model fitness | | | | |
| Log likelihood | -1194.06 | -750.55 | -1036.49 | -722.52 |
| AIC | 2392.13 | 1549.10 | 2098.96 | 1515.06 |
| BIC | 2406.68 | 1712.99 | 2193.53 | 1754.07 |

*p <0.05,

** p< 0.01,

*** p< 0.001.

AIC Akaike's Information Criterion, BIC Bayesian Information Criterion, ICC Intra-Cluster Correlation, PCV Proportional Change in Variance, SE Standard Error

times than those women with no ANC visit(AOR = 2.26; 95% CI:1.12, 4.58, p-value <0.023) and (AOR = 3.34; 95% CI:1.12, 4.58, p-value <0.001) respectively.

The variation of CS delivery also observed across the different regions in-country and residence. The odds CS delivery increased for those women who were living Harari region by 3 times as compared to women who were living Tigray region (AOR = 3.00; 95% CI:1.55, 5.81, p-value <0.001). Regarding residence, the urban women had 2.3 times higher than (AOR = 2.28; 95% CI: 1.35–3.88, p-value <0.002) odds to deliver by CS compared to those women who were living in the rural area.

## Discussion

This study used the EDHS data to analyze the trend of CS rate and identify the factors that associated with CS delivery. The finding of this study indicate that CS rate increased from 0.8% to 6% in 1995–2019 respectively over 24 years of the survey period. This indicate that CS availability has slightly improved. The finding is similar study conducted in Uganda 5.22% [25], in Mozambique 4.7% [17]. Whereas, higher than the study conducted in Egypt 40.1% [26], Nepal 9% [27], and Bangladesh 63% [28]. The low prevalence of CS rate at present study might be due to, the facility and expertise for emergence obstetrics service including CS delivery insufficient and/or sparsely scattered throughout the country; equipment and medicines in the emergency obstetric health centers are insufficient; lack of competent birth attendants and services as well as length of distances and poor landscape without adequate transportation could create a significant geographic barrier to emergency obstetric treatment [29]. Addatinally, lacks information about CS delivery contributing to low acceptance of CS among Ethiopian women, for example, women may be fear of mortality, anxiety about complications, a poor impression of CS as an unnatural technique of delivery, and the expensive expense of the surgery.

The study observed women who were living in the urban area increased CS rate from 5.8% (1995) to 16% (2019) with an absolute change was 10.2%, likewise, women who were living in the rural area CS rate increased from 0.001% (1995) to 3.9% (2015–2016) with absolutely changed 3.8%. Some studies support this finding, a study in Vietnam shows that living in an urban area doubles the likelihood of undergoing a CS [30], and similar findings in Nepal and Mozambique [17, 31], The increasing of CS rate in an urban area compared to that of a rural in Ethiopia might be due to, in rural area have shortage of skilled health workers, high concentration of less experienced doctors and community health workers. Whereas, in urban area the health workforce is disproportionately concentrated, home to all significant private and public health institutions [32] in addition to this, the majority of cesarean births are performed in urban health facilities, notably private clinics [9] although 78.3% of the population lives in rural regions [33].

The finding of this study also indicates a considerable disparity in the service uptake of CS among maternal education, age, parity, body mass index, ANC visit, multiple pregnancy, birth order, and geographical region. Few studies support this finding [16, 26, 31]. This study finding revealed that mothers aged 25 or more had higher odds of delivering by CS compared to women aged less than 25 years old. This finding supported the study in Vietnam [30], Mozambique [17], and Egypt [26]. The reason might be since the risk of delivery problems as well as the chance of premature birth and infant death increased as women got older, excessive bleeding during labor prolonged labor lasting more than 20 hours, and dysfunctional labor that does not progress to the next stage. Furthermore, diabetes and hypertension during pregnancy were more common in older pregnant women, this may lead to increased delivery CS [34, 35].

The odds of women who had completed primary, secondary, or higher education had more likely delivered by cesarean section than women who had not been educated. This finding support study conducted by a global, regional and national representative of 150 countries using secondary data from 1990–2014 [3], similar study from Ethiopia [36], study from Tanzania [37]. The possible reason a higher degree of education enhances the possibility of CS delivery might be women who had received education aware of the costs and benefits of using maternity services and also have more confidence and self-reliance for any decision provided by a health care provider, which in turn, may raise the chance that could get CS service, besides, evidence from health literature demonstrates the fact that health literacy is linked to educational achievement could lead to believe that highly educated women are more likely to seek health services such as CS delivery than their no educated counterparts [29, 38]. ANC visits a significant predictor of cesarean delivery in this study, women who had greater than one ANC visits during pregnancy had more than double likelihood of delivery by CS than those who did have ANC visits. A similar finding reports from a study in Nepal [31] and Nigeria [16]. The possible reason for this could be women get more information about early detection of complications during pregnancy, birth preparedness, complications at each ANC visits. Hence, they may prefer methods of delivery, such as cesarean section. Furthermore, the study also showed, compared to women who had one successful birth, those who had two and four successful births were less likely to undergo a cesarian section similarly, CS delivery declined with increased birth order, those with birth order of greater 4, the odd of having CS delivery was less likely as compared to those with birth order of one. some studies support this finding, including a study in Nigeria [16], a population-based study in Ethiopia [9], Study in Ghana [11]. The possible reason is that women who have not previously given spontaneous successful deliveries have had less experience with methods of delivery and may fear labor pain related to vaginal delivery, thus, this leads to CS delivery.

In this finding, multiple pregnancies were significant factors for CS delivery, the odds of women who deliver twin pregnancies eleven times increased the likelihood of delivery by CS delivery as compared to singleton birth. Few studies support this finding, a study in Nigeria [9], a study in Egypt [27]. The possible reason for this could be multiple pregnancies have been linked to obstetric complications like preterm labor, premature rupture of membranes, malposition, and malpresentation of the fetus. It also increased risks of complications to the mother like gestational diabetes, gestational hypertension, preeclampsia, and intrahepatic cholestasis, all of which may increase the likelihood of giving birth via CS [39, 40]. This study also investigates body mass index greater than 25 as a significant predictor for CS delivery. Women with a body mass index greater than 25 are more likely to deliver by cesarean section as compared to women who had a normal body mass index, little evidence support this finding study in Egypt [26], a population-based study in Ethiopia [10]. The reason for the finding could be obese mothers have a higher risk of pregnancy complications like anemia, hypertension, pre-eclampsia, preterm delivery, emergency cesarean section, and gestational diabetes. Another study also investigated obesity increases the rate of cesarean section, with a rate of 20.7% in a normal weight control group, 33.8% in obese women, and 47.4% in morbidly obese women (BMI>35kg/m2) [41].

This research have some advantages. First, the research was based on big, nationally representative datasets, giving it sufficient statistical power. Second, the study estimates completed after the data had been weighted to allow for probability sampling and nonresponse to make it representative at national and regional levels.

The limitation of this study: for trend analysis the study employed 2019 mini demographic data; however, to assess factors that affecting CS delivery, the study used 2016 data because some factors' data did not complete in the 2019 mini EDHS data.

## Conclusion

In Ethiopia in both urban and rural areas CS rate was low, which did not fall within the WHO's recommended range although a slight improvement observed in the urban area. Factors associated with low prevalence and decrease in the odds of CS delivery included residence in a rural area, lack of ANC visits, and first parity. on the other hand, factors associated with higher prevalence and increased odds of CS delivery included, region, multiple births, maternal overweight, higher educational status, and women of >25 years. The current study reflects unmet needs, which are a known risk factor for increasing maternal and newborn mortality, this highlights the critical need for increased provision and better utilization of life-saving CS in Ethiopia. As result, enhanced availability and access to obstetric care services like CS delivery especially, in rural areas need to be further pursued by meeting the WHO's recommendations.

The research also highlights the importance of addressing the geographic and socioeconomic variables that contribute to Ethiopia's low CS prevalence, thus, empowering women, educating women, implementing health promotion programs aimed at preventing/reducing maternal overweight/obesity, increasing co-services such as ANC usage, and improving maternal awareness could all help to address the current problem.

## Acknowledgments

We'd also want to express our gratitude to the MEASURE DHS Program and ICF International for allowing us to use the EDHS data. We'd like to thank the academic and non-academic staff at the Pan African University Life and Earth Science Institute (PAULESI) at the University of Ibadan in Nigeria.

## Author Contributions

**Conceptualization:** Rahel Mezemir.

**Data curation:** Rahel Mezemir.

**Formal analysis:** Rahel Mezemir, Yadeta Dessie.

**Methodology:** Oladapo Olayemi, Yadeta Dessie.

**Validation:** Oladapo Olayemi, Yadeta Dessie.

**Visualization:** Rahel Mezemir, Oladapo Olayemi, Yadeta Dessie.

**Writing – original draft:** Rahel Mezemir.

**Writing – review & editing:** Oladapo Olayemi, Yadeta Dessie.

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
