## [Decision Letter · Decision Letter 0]

10 Oct 2022

PONE-D-22-08008

Trend and associated factors of cesarean section rate in Ethiopia: evidence from 2000-2019 health survey of Ethiopia

PLOS ONE

Dear Dr. Abebe,

Thank you for submitting your manuscript to PLOS ONE. After careful consideration, we feel that it has merit but does not fully meet PLOS ONE’s publication criteria as it currently stands. Therefore, we invite you to submit a revised version of the manuscript that addresses the points raised during the review process.

Major revisions 

We look forward to receiving your revised manuscript.

Kind regards,

Faisal Abbas, PhD

Academic Editor

PLOS ONE

A clean copy of the edited manuscript (uploaded as the new *manuscript* file).

“Not applicable, since it is secondary data”

“The author declare that they have no competing interest”

6. PLOS requires an ORCID iD for the corresponding author in Editorial Manager on papers submitted after December 6th, 2016. Please ensure that you have an ORCID iD and that it is validated in Editorial Manager. To do this, go to ‘Update my Information’ (in the upper left-hand corner of the main menu), and click on the Fetch/Validate link next to the ORCID field. This will take you to the ORCID site and allow you to create a new iD or authenticate a pre-existing iD in Editorial Manager. Please see the following video for instructions on linking an ORCID iD to your Editorial Manager account: https://www.youtube.com/watch?v=_xcclfuvtxQ.

7. Please amend either the title on the online submission form (via Edit Submission) or the title in the manuscript so that they are identical.

Additional Editor Comments:

major revisions.

Reviewers' comments:

Reviewer's Responses to Questions

**Comments to the Author**

1. Is the manuscript technically sound, and do the data support the conclusions?

Reviewer #1: Yes

Reviewer #2: No

2. Has the statistical analysis been performed appropriately and rigorously? 

Reviewer #1: Yes

Reviewer #2: No

3. Have the authors made all data underlying the findings in their manuscript fully available?

Reviewer #1: Yes

Reviewer #2: No

4. Is the manuscript presented in an intelligible fashion and written in standard English?

Reviewer #1: Yes

Reviewer #2: No

5. Review Comments to the Author

Reviewer #1: 1) i think authors should not mention which software they have used for estimation in abstract.

2) In the last paragraph of introduction, please mention the value addition of the study.

3) Please add another section of literature review after introduction.

4) Why did you use this methodology,method of estimation,please give concrete justification, similarly, why did you select these variables, please explain in detail.

5) Always compare and contrast your findings with the existing literature and elaborate are they similar with the previous studies or there is any difference and how and why?

6) Conclusions, need a serious consideration. Please rewrite it, also add policy implications, limitations of the study and way forward.

Reviewer #2: The paper is very poorly structured. The authors stated that they applied multilevel logistic regression model but I did not found the model in the paper. I did not found any which variables are used at first level, second level and so on. They have not mentioned the how many level variables were used in this study. The authors give unnecessary details of data.

6. PLOS authors have the option to publish the peer review history of their article (what does this mean?). If published, this will include your full peer review and any attached files.

Reviewer #1: No

Reviewer #2: **Yes: **Muhammad Irfan Malik

---

## [Author Response · Author response to Decision Letter 0]

14 Nov 2022

Academic editor

1. Please ensure that your manuscript meets PLOS ONE's style requirements, including those for file naming

Response: Thank you so much for the suggestion; the file's name has been updated to reflect the requirement's style.Hopefully, the style requirements are stick to guideline. 

2. . We suggest you thoroughly copy edit your manuscript for language usage, spelling, and grammar. 

Response: thank you again for your suggestion, the language usage, spelling, and grammar were amended and highlighted with yellow colour on the main manuscript in the file name “support information

Reviewer reports:

Reviewer #1:

1. I think the authors should not mention which software ,they used for estimation in the abstract.

Response:The author’s would like to thank this reviewer for his constructive comments and suggestion. The suggestion was well accepted the authors removed Extended Mantel-Haenszel and STATA 14.1 software from the abstract of the main manuscript

2. In the last paragraph of the introduction, please mention the value addition of the study.

Response:: Again the author’s would like to thank the reviewer for his critical reviewing the relevance of our work, The authors tried to list a few points on the additive value of this study, for instance, the study observing 24-year trend data which highlights national level characteristics.This could lead to policy decisions at the national level and it helps the concerned body that seeks large data (page number 5 and line number 99). Another additive value of this study.included important factors which were not listed in another study that was associated with cesarean section ratelike;,the size of the baby (very large, average, and small )and parity.

3. Please add another section of the literature review after the introduction.

Response: The authors acknowledge the suggestion was given from the review, but it did not meet PLOS ONE's manuscript style requirements. According to PLOS ONE the style ofrequirement literature review is part of the introduction and authors follow this guideline ;thus, the literature review is included in the main manuscript at end of the introduction ( page 4, lines number 72-90).

4. Why did you use this methodology, and method of estimation, please give a concrete justification, similarly, why did you select these variables, please explain in detail.

Response; I would like to thank the reviewer'squestion, it is an important concern. 

Briefly, the study was performed using cross-sectional data obtained from the Ethiopia demographic and health survey (EDHS) conducted in 2000, 2005, 2011,2016, and 2019. These all were nationally representative household surveys conducted by the Central Statistical Agency of Ethiopia. Therefore, the reason for using this methodology in the current study was:

I. The EDHS is the only source of nationally representative household surveys that provide a wide range of data on several health indicators including cesarean sections. 

II. The current study investigates data starting from 1995 to 2019year, which was a 24-year trend of cesarean section rate in Ethiopia,thus, this kind of big data is difficult to access through the primary source data or chart review. Furthermore, this kind of national datais performed meticulously to give sufficient statistical power for analysis. 

III. As we explained previously, the data is big and conducted from eleven regions and two municipal governments of Ethiopia, so all the data is aresummarized in one big figure and making it easy for the reader to look into the whole picture of the problemand to identify the gap.

Response; The reason for the selection of these variables that used in the current study was

firstly, the author search different literature by using the following major database, PUBMED/MEDLINE, HINARI, Google scholar and the key terms used in PubMed search wascesarean section OR abdominal surgery ” AND “ trend OR pattern OR magnitude OR prevalence OR distribution ‘’AND” associated factors “AND “Ethiopia OR Africa OR/ world. after searching all this literature we selected the statistically significant variable. And then the author used two strategies to select these variables:

A. individually choose the independent variable based on their relationship with or influence on the dependent variable 

B. and choose independent variable based on their contribution to the model’s overall performance in explaining the variance of and forecasting the dependent variable 

Finally, before running the final model, multicollinearity was assessed by using the variance inflation factor. Variables highly correlated with other independent variables were excluded from the final multinomial logistic regression model. 

5. Always compare and contrast your findings with the existing literature and elaborate are they similar to the previous studies or if there is any difference and how and why?

Response; The author’s thank the Reviewer for highlighting this point, the author made extensive amendments to the discussion part and saw in detail in the main manuscript. 

6. Conclusions need serious consideration. Please rewrite it, and also add policy implications, limitations of the study, and way forward.

Response; the author rewrite the conclusion and added policy implementation in the main document of the manuscript, but the author put the limitation of study at end of the discussion part(page 30, line number 413-41) because there is no separate heading for the limitation of the study in PLOS ONE manuscript guideline. 

Reviewer #2

1. The paper is very poorly structured. The authors stated that they applied a multilevel logistic regression model but I did not find the model in the paper. I did not find any of which variables are used at the first level, second level, and so on. They have not mentioned how many level variables were used in this study. The authors give unnecessary details of data.

Response; Thank you very much for your constructive question and suggestion, (that shows the model below the description for the question.

The four models were carried out for factors related to CS rate. The first model was an empty model that was used to calculate the random variability in the intercept and there was no variable in this model. The second model estimated the influence of individual-level factors on CS delivery. Variables like; the mother's age, ANC visit educational level, wealth quantile, BMI, parity, and birth order are included in model II ;

The third model looked at how community-level factors are associated with CS delivery. The variable included in model III;are residence and region. 

The fourth model computed the influence of individual and community-level factors on cesarean delivery. The variable included in model IV is;the mother’s age at birth, education, BMI, parity, birth order, ANC visit, type of pregnancy, residence, and geographical region. the detail description (page 24-25), line number 331, ,Table 4. Multilevel multivariable logistic regression output for cesarean section rate among live births in Ethiopia 2016)

---

## [Decision Letter · Decision Letter 1]

17 Jan 2023

PONE-D-22-08008R1Trend and associated factors of cesarean section rate in Ethiopia: evidence from 2000-2019 Ethiopia demographic and health survey dataPLOS ONE

Dear Dr. Abebe,

Thank you for submitting your manuscript to PLOS ONE. After careful consideration, we feel that it has merit but does not fully meet PLOS ONE’s publication criteria as it currently stands. Therefore, we invite you to submit a revised version of the manuscript that addresses the points raised during the review process.

We look forward to receiving your revised manuscript.

Kind regards,

Temesgen Muche Ewunie

Academic Editor

PLOS ONE

Journal Requirements:

Additional Editor Comments:

The authors should revise the figures and tables as per the Journal submission guideline, and try to produce figures with good resolution.

Reviewers' comments:

Reviewer's Responses to Questions

**Comments to the Author**

1. If the authors have adequately addressed your comments raised in a previous round of review and you feel that this manuscript is now acceptable for publication, you may indicate that here to bypass the “Comments to the Author” section, enter your conflict of interest statement in the “Confidential to Editor” section, and submit your "Accept" recommendation.

Reviewer #1: (No Response)

Reviewer #2: All comments have been addressed

2. Is the manuscript technically sound, and do the data support the conclusions?

Reviewer #1: (No Response)

Reviewer #2: Yes

3. Has the statistical analysis been performed appropriately and rigorously? 

Reviewer #1: (No Response)

Reviewer #2: Yes

4. Have the authors made all data underlying the findings in their manuscript fully available?

Reviewer #1: (No Response)

Reviewer #2: Yes

5. Is the manuscript presented in an intelligible fashion and written in standard English?

Reviewer #1: (No Response)

Reviewer #2: Yes

6. Review Comments to the Author

Reviewer #1: (No Response)

Reviewer #2: The authors explore an import social issues of our society. The C-section is very common issue in developing countries. I really enjoyed the reading of this paper and I have some issues with the writeup. Authors have comprehensively addressed all the issues raised by me on previous draft of the paper.

7. PLOS authors have the option to publish the peer review history of their article (what does this mean?). If published, this will include your full peer review and any attached files.

Reviewer #1: No

Reviewer #2: No

---

## [Author Response · Author response to Decision Letter 1]

26 Feb 2023

Rebuttal letter 

Manuscript ID: D-22-08008

Manuscript Title: Trend and associated factors of cesarean section rate in Ethiopia: evidence from 2000-2019 health survey of Ethiopia

I thank again the editor and two reviewers very much for their response and suggestions on tour manuscript. To show our response to each point raised by the editor as below.

The authors included the response within the response to reviewers' box in the submission system and modifications are highlighted to changes the original version of the main manuscript.

 The authors addressed all the issues raised by the Editor. I hope that the response satisfied you all. The revisions significantly improved the manuscript. 

Sincerely 

On behalf of all the authors 

 Rahel Mezemir.

Response to comments 

1. The authors should revise the figures and tables as per the Journal submission guideline, and try to produce figures with good resolution

Response: The authors would like to thank the editors for their constructive comments and suggestions. The suggestion has well accepted the modification of the table and figure made according to PLoS one journal requirements.

2. Concerning the reference list, the authors changed the reference style from Vancouver to PLoS style in a word document in the menu of the reference style list .The authors did not use retracted paper in the reference list.

---

## [Editor Report · Decision Letter 2]

28 Feb 2023

Trend and associated factors of cesarean section rate in Ethiopia: evidence from 2000-2019 Ethiopia demographic and health survey data

PONE-D-22-08008R2

Dear Dr. Abebe,

We’re pleased to inform you that your manuscript has been judged scientifically suitable for publication and will be formally accepted for publication once it meets all outstanding technical requirements.

Kind regards,

Temesgen Muche Ewunie

Academic Editor

PLOS ONE
---

## [Editor Report · Acceptance letter]

6 Mar 2023

PONE-D-22-08008R2 

Trend and associated factors of cesarean section rate in Ethiopia: evidence from 2000-2019 Ethiopia demographic and health survey data 

Dear Dr. Mezemir:

I'm pleased to inform you that your manuscript has been deemed suitable for publication in PLOS ONE. Congratulations! Your manuscript is now with our production department. 

Kind regards, 

on behalf of

Mr. Temesgen Muche Ewunie 

Academic Editor

PLOS ONE